# From Identity to Ambugity: Exploring Interprofessional Collaboration Opportunities for Pharmacists in Rural and Remote Australia

**DOI:** 10.3390/pharmacy11020077

**Published:** 2023-04-20

**Authors:** Selina Taylor, Alannah Franich, Sophie Jones, Beverley D. Glass

**Affiliations:** 1Pharmacy, College of Medicine and Dentistry, James Cook University, Townsville, QLD 4811, Australia; 2Murtupuni Mount Isa Centre for Rural and Remote Health, Mount Isa, QLD 4825, Australia

**Keywords:** exercise physiologist, podiatrist, role theory, interdisciplinary

## Abstract

Rural and remote populations are predisposed to poorer health outcomes, largely associated with limited access to health services and health professionals. This disparity provides an opportunity for health professionals to work collaboratively in interdisciplinary teams to deliver improved health outcomes for rural and remote communities. This study aims to explore exercise physiologist and podiatrist perceptions of interprofessional practice opportunities with pharmacists. Role theory provided a framework for this qualitative study. Interviews were conducted, recorded, transcribed, and thematically analysed according to the constructs of role theory (role identity, role sufficiency, role overload, role conflict, and role ambiguity). The perceptions of participants varied, largely due to the lack of understanding of the role and scope of the practice of a pharmacist. Participants acknowledged and adopted a flexible approach to the way in which they delivered health services to meet the needs of the community. They also described a more “generalist” approach to care, owing to the high prevalence of disease and disease complexity, along with a lack of staffing and resources. The potential for increased interprofessional collaboration was supported and identified as a strategy to manage significant workloads and provide improved patient healthcare. The application of role theory to this qualitative study provides insight into perceptions of interprofessional practice that may inform future development of remote practice models of care.

## 1. Introduction

Australia’s rural and remote populations experience a 20% higher disease prevalence and shorter lifespan in contrast to metropolitan populations, due to disparities in social determinants and inequitable access to comprehensive healthcare [1,2,3]. This is attributed to a large geographic spread, low population density, poor health literacy, limited funding, and insufficient infrastructure [1,2,3]. Rural and remote populations are equipped with limited health services, referral pathways, and specialist care, resulting in the current health demands of the communities not being met [1,2,3]. Low health literacy is a major barrier to healthcare, which has complexities associated with other determinants of health, including socioeconomic status and education, physical environment, support network, gender, genetics, and ethnicity [4].

Interdisciplinary healthcare teams can provide effective, affordable, and accessible healthcare to optimise health outcomes of patients within rural and remote communities [5]. Interprofessional collaboration provides improved healthcare provision through the delivery of expanded skill sets, which allows for the most appropriate healthcare provision for patients [5]. However, to successfully implement interdisciplinary healthcare teams, where collaboration occurs between healthcare professionals from different health domains, potential barriers preventing current implementation must be addressed [6].

Due to the scarcity of health professionals and readily available resources, health professionals utilise compensatory pathways of health delivery, which are extensively time consuming, labour intensive, and less effective, generating suboptimal patient health outcomes [6]. The Australian Institute of Health and Welfare has highlighted the importance of improved access to Australia’s health workforce regarding the distribution of health practitioners within rural and remote communities [7]. It has been recognised that the number of full-time employees working within their registered professions decreases with increasing remoteness of practice [7]. There are also more registered clinical full-time health professionals in major cities than within all regional and remote areas of Australia combined [7]. Ultimately, this maldistribution contributes to significant health disparities between rural populations and their metropolitan counterparts [3].

Evidence indicates interdisciplinary teams involving allied health professionals and pharmacists deliver improved patient health outcomes, resulting from the health professional’s scope of practice, knowledge, and work experience [8]. Interprofessional collaboration has demonstrated a reduction in errors, adverse events and length of hospital admissions, enhanced staff communication, and both patient and staff satisfaction [8]. Pharmacists are highly trusted and accessible healthcare providers, currently ranked the third most trusted profession within Australia [9,10]. The role of pharmacists is evolving, with expansion of their roles emerging to address the current lack of services and access to health professionals [11,12].

Evidence of interprofessional collaboration between exercise physiologists and pharmacists has highlighted improved patient health outcomes through the prevention and improvement of cardiovascular health and prevention of injuries and fractures associated with osteoporosis [13,14,15]. Exercise physiologists create and implement personalised exercise programs and coaching for patients [14]. The pharmacist’s role has included tasks such as screening patients’ blood pressure, cholesterol, and glucose levels and undertaking weight and waist circumference measurements. Pharmacists have been integral in educating patients on their disease state, along with dispensing, supplying, and monitoring patients’ medications [13].

Healthcare provided through collaboration between podiatrists and pharmacists has also improved the quality of life for patients experiencing foot-related disorders and diabetes [16,17]. Pharmacist undertaking tasks including testing blood glucose levels, counselling, and educating patients about health conditions and medications concurrently with podiatrists performing regular foot examinations and providing education on the use of diabetic socks resulted in a reduced prevalence of diabetic foot infections [16,17].

These multidisciplinary teams enhance staff communication, reduce errors that result in adverse events and length of hospital stays, and enhance both patient and staff satisfaction [8]. Healthcare workers included in interdisciplinary teams develop an extended scope of practice to be applied to achieve improved health outcomes for patients [18]. This finding is also true for the collaboration between pharmacists and other allied health professions, facilitating the improvement of patient health outcomes through increased access and reduced costs of healthcare [8,19]. This collaboration is, thus, essential to achieving improvement of the health of rural communities [8,19].

Previous studies have identified that the broad implementation of interdisciplinary healthcare is challenged by factors associated with the constructs of role theory, which consists of role identity, role sufficiency, role overload, role conflict, and role ambiguity [20,21]. The aim of this study was to explore these constructs to understand the attitudes and perspectives of allied health professionals regarding their own professional practice and interprofessional collaboration opportunities between exercise physiologists and podiatrists, with pharmacists in rural and remote Australia.

## 2. Materials and Methods

### 2.1. Study Design

In-depth, semi-structured interviews were conducted with rural and remote exercise physiologists and podiatrists, with the application of an ethnographic lens by immersing researchers within rural and remote communities to enhance this phenomenological descriptive, qualitative study.

### 2.2. Participants, Setting and Recruitment

A total of 14 participants took part in this study, with the condition of participation including the need to have experience working within a rural or remote community, be freely willing to partake in the study, and provide consent to allow the information they provided to be used in the study. To recruit participants, during February 2022, rural exercise physiologists and podiatrists were contacted via email and/or phone to discuss their potential interest regarding participation in the study. Recruitment through the pandemic was difficult; many participants agreed to be involved in the study on the condition the interviews occur online (via zoom), while few participants requested the interviews to be conducted in person, in Mount Isa, Queensland, Australia.

### 2.3. Data Collection

Allied health professionals who agreed to partake in the study were provided study details and written consent forms to confirm their participation. Interviews were scheduled and carried out according to the preferences of the participants (face-to-face and online). An interview guide developed in accordance with Hardy and Conway was used to semi-structure the interview process (Appendix A) [18]. Participant demographics were collected as well as occupation, years of practice (both in urban and rural settings), and motivation for rural practice.

### 2.4. Data Analysis

All interviews were manually transcribed verbatim which allowed researchers to familiarise themselves with the data in context of the research. All identifying data were removed. Data were thematically analysed in accordance with Braun and Clark to identify the emerging themes of role theory [22]. All interviews were analysed, and relevant data were categorised into the five constructs of role theory by the interviewer before being revised by co-researchers. This revision of the analysis by co-researchers reduced misinterpretation and the manipulation of the data. Bias and assumed knowledge were minimised through a member-checking validation process.

### 2.5. Ethics Approval

Ethical approval for this study was granted by the James Cook University Human Research Ethics Committee (H8278).

## 3. Results

Fourteen health professionals participated in the study, seven exercise physiologists, and seven podiatrists. All participants were in a rural location as defined by the Modified Monash Model (MMM) rating between 4 and 7 [23]. COVID-19 influenced the method of data collection and, thus, eleven interviews were conducted online and three were face-to-face. Two-thirds of the participants were aged between 25 and 35 years. Participant demographics are summarised in Table 1.

The interview transcripts have been analysed using the role theory framework, consisting of five role constructs defined in Table 2 [24]. These constructs refer to cultural norms in which an individual holding a certain role in society is expected to follow, outlining expectations surrounding one’s behaviour, duties, and identity. The constructs of role theory have enabled the identification of factors influencing poorer health outcomes in rural and remote communities and the exploration of the perceptions of allied health professionals regarding their views on interdisciplinary healthcare delivery. Direct quotes from participants are annotated with participant occupation and an identifying numeral.

### 3.1. Role Identity

Exploration of role identity revealed that all rural allied health participants identified as rural generalists, due to working with a variety of presentations requiring complex care regimens. As a result of the variety of presenting conditions, these professionals in rural and remote Australia are required to implement innovative and adaptive practice techniques to accommodate for the unpredictability of the clientele and tasks.


*“You do have to try to take on more of a generalist approach, while still staying within your scope of practice … you have to be very adaptable and creative in how you might approach your care.”*
(Exercise physiologist 7)

It was agreed that the development of these innovative approaches to healthcare is unique to rural and remote healthcare. Participants recognised that they were presented with and were able to treat patients with conditions, which they would likely not be exposed to in an urban practice. Adopting a generalist approach to managing for many participants was an enjoyable aspect to working in rural communities and posed an opportunity to “*do a bit of everything*”.


*“I see a lot of different conditions that come through the clinic daily that I probably wouldn’t see back home, or wouldn’t have the opportunity to see being a new grad.”*
(Exercise physiologist 2)

Participants agreed that within metropolitan locations, where healthcare is accessible, allied health is often compartmentalised and restricted to working within a specific field, armed with a narrow or specific scope, with the inability or inexperience to adapt to specific patient needs [25]. In comparison, rural practice offers them the opportunity to expand the skills and scope of their practice, resulting in improved comprehensive patient care [12].


*“I feel like you’re not just a podiatrist when you’re working in rural healthcare, there’s a lot more, you become a psychologist and you become a social worker… there’s so much more to it than just being a podiatrist”.*
(Podiatrist 1)

### 3.2. Role Sufficiency

Participants supported the notion that rural health professionals are required to be versatile, often practising with a broader scope to effectively provide and meet the needs of patients [2]. There was consensus amongst participants that rural allied health professionals were more under-resourced than their metropolitan counterparts. Participants were unsatisfied with the lack of available resources and equipment, which is largely associated with a lack of funding, essentially leaving practitioners unable to sufficiently fulfil their roles as exercise physiologists and podiatrists to support their communities.


*“I did some calculations…the total number of visits that I was supported to go to that community was inadequate to meet the bare minimum of the national guidelines.”*
(Podiatrist 2)

Most participants agreed that they were resourced to deliver simple, non-complex tasks; however, they were insufficiently resourced both in the context of healthcare staffing and equipment to provide sufficient care for complex patients.


*“I realised that, sometimes I was their only contact of care and I’d have to adopt more of a generalist approach to seeing these patients with limited basic resources available.”*
(Exercise physiologist 7)

It was suggested that increased, sufficient access to reliable and improved resources would have a positive impact on the health professional’s ability to work to optimise health outcomes for patients in rural and remote communities. A significant barrier to this taking place was identified as the absence of leadership and lack of community-centred care and cultural sensitivity, due to short-term, frequently changing practitioners working within these rural communities [26].


*“I think the hardest part with working in remote areas like Mt Isa is that you don’t get that consistency in leadership and that consistency in leadership is what drives projects and resources from year to year.”*
(Exercise physiologist 6)

### 3.3. Role Overload

Allied health professionals expressed that an extensive workload was placed upon them whilst working in rural and remote communities of Australia. Overload experienced by health professionals was described as being further exacerbated by extensive travel distances, poor health literacy, and considerably lower socioeconomics generally held by rural populations. Participants expressed that they felt they carried a heavy workload, having to constantly take on more clients, service outreach communities, and work ten (or more) hour days, while still having patient waitlist times of up to four months. Participants acknowledged feelings of helplessness, burnout, and dissatisfaction associated with being unable to provide communities with adequate healthcare.


*“I always felt as though, no matter how hard I worked. It was never enough to meet the need.”*
(Podiatrist 2)

It was also agreed that, similar to what is reported in the literature, staff recruitment and retention is difficult, contributing to the overload faced by health professionals already working in rural and remote areas [27]. New graduates reported experiencing uncertainty about practising within rural and remote communities due to the professional isolation, lack of professional incentives, community-cultural awareness, and peer support. Conversely, the attraction for health professionals to work in rural areas was reported to be generous pay offers. However, as a result, often practitioners will go to these locations for short periods of time before relocating. The positive impact of locally employed health professionals was also highlighted.


*“…most of us just come here to have a bit of a stint and then we go, and the next person comes in, so I think that’s a huge challenge here is keeping people here.”*
(Exercise physiologist 5)


*“It builds a lot more confidence with patients when you actually live here …it allows a rapport to be built … living here just offers that ability of being able to look after your patients better.”*
(Podiatrist 4)

### 3.4. Role Conflict

It was agreed that often little to no interprofessional conflict occurs within the under-resourced rural and remote communities, mostly due to the informal collaborative partnerships already in place. These collaborative care approaches have developed as a result of the physical proximity of allied health professionals, similar social groups, broadened scope of practice, and complex patient loads. Despite conflict being reported as minimal, it was apparent that newer allied health practitioners, who were unfamiliar with the demands and needs of rural communities, may find themselves contributing to conflicting situations.


*“There’s definitely times that we overlap. Sometimes people sort of instead of just giving it to me to speak about it, they sort of try and give their own opinions on stuff. Even if they do not have a degree in the area but anyway. It does help when you work in a multi-disciplinary team, and everyone knows what you do.”*
(Exercise physiologist 1)

Participants welcomed the idea of increased and improved collaboration that would allow for reduced workloads, prevention of duplicated therapy, provision of opportunities for continual professional development, and, most importantly, improved patient health outcomes. Participants predominately shared no concerns of having others over-stepping professional boundaries related to the scope of practice. They encouraged the notion of shared knowledge from their specialised areas interest to the wider community’s health [28]. Participants expressed that territorial behaviour regarding knowledge and available services was single-minded and resulted in the creation of a larger workload for oneself.


*“… it’s really important rurally to be able to have that crossover …I would encourage it, I’d love it if people wanted to massage and needle people and get them moving, and they’re willing to do it and they’re a “speachie”, I don’t mind.”*
(Exercise physiologist 6)

Participants were supportive of a crossover of the scope of practice between allied health professionals, as it enabled flexibility of care to be delivered to rural and remote communities that do not have constant access to specialised health care professionals. In addition, participants expected a continuation of care to be delivered and available to rural and remote populations, particularly benefiting patients with complex disease states.


*“There were people who were doing diabetic foot checks while we weren’t there, people doing wounds dressings and cutting toenails for people in aged care facilities. All these things found within the scope and area of expertise with podiatrists. But we encouraged other health professionals to get an understanding of what we do and upskilling in those areas to complement the work that we did.”*
(Podiatrist 2)

### 3.5. Role Ambiguity

A lack of knowledge of different health professional roles and skills, particularly for recently-graduated allied health professionals, contributed to an increased workload for other health professionals, resulting in excessive wait times and, therefore, preventing patient health optimisation. The role of the pharmacists was recognised as not limited to the supply of medicines but including checking for drug interactions, medication safety, and educating patients [29]. However, some recently-graduated practitioners were not able to comprehend the benefits of working alongside a pharmacist or understand the benefits of the role of the pharmacist in multidisciplinary teams. A notion of referring patients with a medication-related issue was identified by two participants.


*“I don’t know if this is because of the way we’ve been taught but the GP is like the central hub. So you’d always go back to the GP and the GP is the one that refers to who needs to be involved in the care. Even during uni I wouldn’t say we learnt too much about working together with a pharmacist.”*
(Exercise physiologist 5)

Participants discussed their views regarding collaborative integration with pharmacists as positive, with the potential outcome of improved patient health [30]. Most participants viewed the implementation of interdisciplinary teams, especially for rural and remote Australia, to be an appropriate and feasible solution to existing barriers. However, a recurring concern raised was the lack of understanding of other allied health professional’s roles. Exercise physiologists expressed their concerns regarding their lack of recognition as a health professional in rural Australia. The exercise physiologist role is relatively new and often patients, and other health professionals, *“are not aware”* of the benefits an exercise physiologist can have on a patient’s overall health outcome. This lack of knowledge surrounding the role of other health professionals ultimately leads to reduced referrals in an environment already lacking referral pathways and specialised services [31].


*“Most of the time people don’t actually understand what our role is in the allied health field.”*
(Exercise physiologist 2)

Largely, increased clinical experience correlated with an increased recognition of the potential opportunity for pharmacist collaboration and inclusion within interdisciplinary health teams.


*“Many patients are on various medications that, and I use this term very loosely, “stabilise” them until they’re able to see a specialist. That can definitely reduce an exercise physiologist’s workload, or any other allied health members workload because that’s one less thing we have to go and dig into and then we can look at another patient sooner.”*
(Exercise physiologist 7)

Participants who had previously experienced collaboration with a pharmacist reported positive outcomes occurring, including more enjoyable work, reduced workloads, improved interprofessional relations, and improved patient health outcomes. These participants were optimistic regarding the future and potential for increased, widespread interprofessional collaboration between themselves, other allied health professionals, and pharmacists.


*“I definitely think interprofessional collaboration is important… you’re giving the patient the best outcome for whatever condition or rehab that they’re going through… they’ll be able to have overall the best outcomes.”*
(Exercise physiologist 2)

Those participants who had a lack of understanding of the role of a pharmacist had difficulty considering the benefits associated with pharmacist collaboration for themselves or their patients.


*“No, I don’t see any point in collaboration…You know, you’re giving out medications…The days are long gone where you’re counting out 10 pills and putting them in a bottle. No, you just have to label it and make sure you put the label on the right box.”*
(Podiatrist 3)

Conversely, participants with experience in interdisciplinary teams who had a thorough understanding of the pharmacist’s role were able to recognise the benefits of collaboration.


*“…the scope of practice that pharmacists have, I think that needs to be shared more because it’s just not the drug dealing … it’s a lot more to it than that.”*
(Exercise physiologist 6)


*“The pharmacist, I guess, in basic terms is that they are getting the scripts in. To start off with, making sure that the medications that they’re providing aren’t contraindicated with any of the other medications that they are dispensing, making sure that the clients also have access to other health professionals and look at them as a holistic person.”*
(Podiatrist 5)

## 4. Discussion

Conflicting perceptions surround the potential collaboration between allied health professionals, including exercise physiologists, podiatrists, and pharmacists. This manifests as either perceived eagerness or reluctance attributed to the existing knowledge of the participants of the role of the pharmacist. Reluctance towards interprofessional collaboration has resulted in an increased workload, due to a lack of referral pathways and the utilisation of existing allied health professionals. Consequently, most participants identified as rural generalists, providing varied health services to many patients. Amongst allied health professionals within rural communities, minimal conflict occurred due to the lack of competitive practice and the proximity of a limited number of allied health professionals available for collaborative patient care.

Role identity relates to the anticipated knowledge of an individual’s role and corresponding scope [24]. The scope of practice of rural and remote allied health professionals is broad and associated with the significant variation and complexity in patient presentations and complexity [32]. This broad scope is often referred to as a generalist approach, naming those practitioners working remotely “rural generalists” [32]. The rural generalists have arisen due to inequitable access to affordable healthcare. These practitioners have provided comprehensive, enhanced primary healthcare to the rural communities, enabled via the possession of unique skill sets, innovative practice techniques, and a broad knowledge regarding an array of conditions, accommodating for most daily presentations [33,34]. Rural generalists, thus, present a great advantage to rural and remote communities, enabling barriers of access to be reduced [8,18,35]. When health practitioners have a lack of social-ecological, cultural involvement and understanding, a patient’s involvement and views surrounding their healthcare may be significantly impacted [35]. The involvement of health practitioners within the local community is associated with improved health outcomes and health productivity amongst rural populations [36]. Immersing health practitioners within the community enables poor health literacy to be addressed, optimising health outcomes due to improved active engagement in healthcare [36].

The appropriate distribution of health practitioners, scope, knowledge, and resources to fulfil assigned clinical duties and expectations is encompassed in the construct ‘role sufficiency’ [24]. Currently, there is a maldistribution of health staff and resources to service Australia’s rural and remote communities [3]. In 2020, it was found that major Australian cities were equipped with almost three times (×2.92) the number of full-time health employees when compared to the entirety of Australia’s rural and remote communities [7]. This study identified a lack of resources compared to urban counterparts; this is attributed to the lack of skilled staff, physical resources, and equipment, along with funding, lack of community, cultural awareness, and leadership. These barriers, which prevent the needs of the community from being met, have been repeatedly identified as significantly limiting the delivery of healthcare [37,38,39]. Overall, despite rural health professionals working with limited resources, they effectively utilise an adaptive approach to practice. Rural practitioners held optimistic views towards continuing to develop increasingly innovative approaches to healthcare delivery to better service their community’s health needs.

Role overload is associated with workplace demands placed upon an individual, exceeding personal and professional ability [24]. Rural health services have significantly larger workloads placed upon them in contrast to those in urban locations, as evidenced by the extensive work hours and wait times experienced by patients ranging from weeks to months [3,40]. This is exacerbated by the increased prevalence of chronic disease, along with insufficient resources and staff maldistribution [3,40]. This study identified that the overload experienced by rural health professionals is further challenged by excessive travel distances to health facilities, poor health literacy, education, and lower socioeconomics [41]. Data show that, as the COVID-19 pandemic impact continues, an associated increased incidence of illness follows, further contributing to rural health professional overload [7]. This study also identified that health practitioners working within rural and remote regions of Australia felt helplessness, burnout, and dissatisfaction with their job [42]. To address some of the existing issues, in recent years, incentives to work and stay within rural communities have been offered, including higher wages, accommodations, and funding for travel [43]. Despite these efforts, in addition to the fact that over 60% of participants had worked rurally for fewer than five years, health professionals’ motivation to work rurally long-term is lacking. A strategy trialed by another study explored the implementation of interdisciplinary teams within a healthcare setting for six months [44]. The study found that the implementation of the interdisciplinary team was associated with increased job satisfaction, reduced staff turnover, and improved communication between staff and patient health outcomes associated with the increased accessibility to healthcare and an improved continuum of care [44,45].

Role conflict is common in all workplace settings and refers to obstacles preventing the completion of duties to the best of a health professional’s ability [24]. The participants of this study were encouraged by interprofessional collaboration to minimise discrete role conflict. They supported the idea of increased collaboration as it would allow for the continuation of professional development, reduced workloads, prevention of duplicate therapies, improved job satisfaction, and patient health outcomes [3,8,46]. A broadened scope of practice was suggested to improve care for residents who lack consistent access to health professionals, leading to reduced health costs for patients, more sustainable healthcare outcomes, and job satisfaction [8,19,46,47]. Enhanced patient care through optimised drug management and better clinical documentation, services, and recommendations has been attributed to the inclusion of pharmacists within interdisciplinary teams [48]. This study confirms that health professionals working in rural areas would expect similar outcomes.

Role ambiguity occurs when uncertainty or confusion develops due to a lack of understanding regarding the role of other health professionals [24]. Podiatrists and exercise physiologists were interested in considering the idea of interprofessional collaboration with pharmacists. Throughout this study, it was identified that those allied health professionals who had a limited understanding regarding the role of other allied health professionals also lacked previous experience in an interdisciplinary team. This highlights a major barrier to current adequate healthcare delivery to rural areas. While the concept of the “rural generalist” is well recognised in rural practice in Australia, the potential for interprofessional practice between allied health professionals and pharmacists is less accepted. This is probably due to a lack of understanding of the role of the pharmacist, as highlighted in the literature, which reports that a clear understanding of one’s role is one of the many contributing factors preventing interprofessional collaboration. Therefore, it is recommended to introduce and educate health students early on the importance of interprofessional collaboration and the role of other health professionals [49,50,51]. In order to optimise health outcomes and limited resources within rural communities, these interdisciplinary teams should be created with an understanding of each other’s roles.

Limitations of this study are associated with a small sample size and limited geographical distribution, impacting the generalisability of the results to all rural health professionals. In addition, due to COVID-19 precautions, most interviews were conducted via online platforms.

## 5. Conclusions

Rural health practitioners are faced with increasing demands for healthcare associated with the lack of availability and access to healthcare professionals and resources, resulting in increased workloads. Health professionals are less available in rural and remote communities and, therefore, are required to be increasingly versatile due to the large variety and complexity of presentations. Currently, pharmacists are underutilised despite their specialist knowledge of medication and their expanding scope of practice. Collaboration in rural and remote communities presents as an appropriate and feasible solution to address the barriers concerning inadequate access and lack of healthcare services within these communities. Allied health practitioners are welcoming of implementing an interdisciplinary healthcare approach and have identified positive outcomes for both the patients and health professionals. Future models of care for rural and remote practice would benefit from focused support for understanding the role of all health professionals included in a team.

## Figures and Tables

**Table 1 pharmacy-11-00077-t001:** Participant Demographics.

Participant Characteristics		Total (%)	
Gender		Age	
Female	6 (42.9%)	Under 25 years	3 (21.4%)
Male	8 (57.1%)	25–35 years	9 (64.3%)
		Over 35 years	2 (14.3%)
**Occupation**		**Total years of practice**	
Exercise physiologist	7 (50%)	Under 5 years	5 (35.7%)
Podiatrist	7 (50%)	5–10 years	6 (42.9%)
		More than 10 years	3 (21.4%)
**Years of rural practice**		**Years of urban practice**	
Nil years	0 (0%)	Nil years	8 (57.1%)
Under 5 years	9 (64.3%)	Under 5 years	5 (35.7%)
5–10 years	2 (14.3%)	5–10 years	1 (7.2%)
More than 10 years	3 (21.4%)	More than 10 years	0 (0%)

Table Legend—N = 14.

**Table 2 pharmacy-11-00077-t002:** Definitions of the constructs of role theory [24].

Constructs	Definition
Role identity	Anticipated role and knowledge expected of an individual in a specific social position.
Role sufficiency	Appropriate scope of practice and knowledge allocated to complete clinical duties.
Role overload	Workplace demands placed on an individual exceeds personal/professional resources.
Role conflict	Obstacles preventing duties from being completed to the best of an individual’s ability.
Role ambiguity	Uncertainty or confusion emerging due to vagueness of the role of another health professional.

## Data Availability

Data is unavailable due to privacy restrictions.

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
