# Peer review of "From Identity to Ambugity: Exploring Interprofessional Collaboration Opportunities for Pharmacists in Rural and Remote Australia"

_pharmacy, 2023, doi:10.3390/pharmacy11020077_

Round 1

Reviewer 1 Report

Thank you for this paper. It is very well written and constructed.

The opportunity for inter professional collaborations in the community and particularly in regional and rural areas is important for pharmacy. There was a slight misalignment for me between the title and the text. while some of the introduction pointed to the importance of the relationships and IPE options for pharmacists, the methods and the subsequent results didn't immediately point to the pharmacists / health professional connection. It was interesting to read about pod and EP's though not specifically clear why they were chosen over other HP in the rural and remote context -- e.g. Social workers, OT, PT, Psych? The SSI outline only has a "prompt" on pharmacists in the questions under role ambiguity. Some of these things may influence the outcomes and experience. EP would not usually have much exposure in their work or training to pharmacists, where Pod might. So, I wonder how this really interesting piece around identity and ambiguity can be tightened to realign to the question of how pharmacists can engage more in the IP team and maybe a tweak to the title and intro could get the flow and discussion working more closely.

There are good sessions to be learnt here. I think that there are very valuable commentaries here not just for the R+R work but broadly around the impact that is felt of identity, ambiguity and conflict are particularly relevant. IPE in the context of training and development in university and the understanding of role and contribution was highlighted in this document. Questions of "scope" - what is scope for a profession, potential scope or even expected in a particular team is relevant in urban settings too and variable between individuals. There will always be a "profession scope" and then the individual within ....

I think that this is a piece of work of value that could benefit from a think about how the structure better represents the data collected. It is well written and otherwise presented.

Author Response

Thank you for your review, your comments are appreciated.

Points from your report that I have addressed.

There was a slight misalignment for me between the title and the text. - thank you - title has been reworded.

It was interesting to read about pod and EP's though not specifically clear why they were chosen over other HP in the rural and remote context -- e.g. Social workers, OT, PT, Psych?  - HPs were chosen on availability.

The SSI outline only has a "prompt" on pharmacists in the questions under role ambiguity. Some of these things may influence the outcomes and experience. - thank you - the interview guide was designed to understand participants existing perceptions around the pharmacists role and it highlighted that as you state, may EPs and Pod may not be aware of the IP possibilities for practice. Future studies are planned to explore this area more deeply to examine future opportunities in practice. 

Maybe a tweak to the title and intro could get the flow and discussion working more closely - thank you - minor edits have been made to title and  intro.

Thanks kindly

Reviewer 2 Report

Thank you for the opportunity to review this manuscript.  This paper examined interprofessional collaboration for pharmacists in rural and remote Australia.  The results showed the insights and practices of current rural allied health practitioners, including their views on interdisciplinary teams. The results of this study appear to largely be in agreement with, but have extended upon, the existing literature on this topic. 

The manuscript is well written, the objectives were clear and methodology sound.  I found the results quite interesting and well presented, although I do feel overall the authors could bring more of a connection to pharmacy/pharmacists/pharmacy practice context with all the constructs (given that this is a pharmacy journal), rather than just with the role ambiguity construct seeing that link.  Similarly, I would have liked to see more of a link in the discussion of this study to the context of pharmacy and synthesising some more insights on this.

The limitations and recommendations were appropriate, with some important conclusions made and some lessons that can be learnt.  Overall, I enjoyed reading this manuscript and learning the findings.  I just have a few minor comments.

Line 104: and/or

Line 108:  For clarity, it is best to describe that Mt Isa is in Queensland, Australia

Line 136: I would suggest inserting ‘n=14’ for clarity, either in table legend or within the table

Line 316: Minor point - Missing dash before (Exercise physiologist 6)

Author Response

Thank you for your review, your comments are appreciated. We are hoping to conduct further research in this area and explore the area with more depth in future. 

Line 104: and/or - corrected - thank you

Line 108:  For clarity, it is best to describe that Mt Isa is in Queensland, Australia - corrected thank you.

Line 136: I would suggest inserting ‘n=14’ for clarity, either in table legend or within the table - added - thank you.

Line 316: Minor point - Missing dash before (Exercise physiologist 6) - included thank you.